# Cryptic biodiversity of freshwater fish species in Bangladesh

**Mahmudul Hasan**[1,2*], **Chiaki Kambayashi**[3], **Zahid Hasan Anik**[2], **Md. Saiful Islam**[1]

**1** Department of Fisheries, Jamalpur Science and Technology University, Jamalpur, Bangladesh,
**2** Evolution and Diversity Research Laboratory, Jamalpur Science and Technology University, Jamalpur, Bangladesh, **3** Faculty of Science, Niigata University, Niigata, Japan

* mhasan@jstu.ac.bd

## Abstract

Unrecognized cryptic species impede conservation planning and biodiversity assessments. DNA barcoding has tremendously expanded the number of novel and cryptic species in biological science. Despite few sporadic studies, the exact number of freshwater species found in Bangladesh is not known. To assess this biodiversity, we sequenced the COI gene of 124 freshwater specimens, which were gathered from various localities around Bangladesh. Seven cryptic species hidden among the currently studied specimens were identified based on the findings of phylogenetic and species delimitation analyses. The preliminary assessment also encompassed a restricted morphological examination of these cryptic taxa. The appearance of cryptic species, some of them possibly endemic, has been hypothesized. This raises concerns regarding the true diversity and evolutionary history of freshwater species in Bangladesh, which are significantly underrepresented in the current systematic frameworks that do not account for DNA data. Our current study provides baseline data that might aid local ichthyologists in their quest to identify additional new species by combining several variables (morphology and ecology). Further research is warranted to protect the priceless freshwater species in Bangladesh.

## Introduction

Bangladesh is situated in the oriental region of the Indo-Himalayan and Indo-Chinese sub-regions [1]. This country is home to the Ganges-Brahmaputra river delta, which is the largest delta in the world. Approximately 230 rivers crisscross through the lands of Bangladesh. Among these rivers, 57 are transboundary rivers (54 rivers are shared with India and three are shared with Myanmar) [2,3]. The downstream region consisting of many different rivers, canals, lakes with both flowing as well as static water (beels), and harbors are home to several freshwater fish. Approximately 265 species of freshwater fish are found in this lowland nation [4]. Freshwater fish are found in Bangladesh's biggest river systems, the Ganges, Brahmaputra, and Meghna, which are collectively equal to 4,339,694 hectares of inland water [5]. Approximately 30.2% of all fish species found worldwide are freshwater fish [6] and these are very valuable both nutritionally and economically [7]. The fishing industry is unquestionably important for Bangladesh's economy, culture, and nutrition [8,9].

**Data availability statement:** All DNA sequences files are available from the GenBank

database (accession numbers: LC823243 and LC823366). Other relevant data are within the paper and its supporting files.

**Funding:** The author(s) received no specific funding for this work.

**Competing interests:** The authors have declared that no competing interests exist.

According to Shelley et al. [10], cryptic fish species present one of the main obstacles to the proper management, conservation, and planning of fish biodiversity. Cryptic species, of which are genetic variants, however, physically identical are a common occurrence and they are concealed as nominal species. Although morphology-based investigations have been conducted in Bangladesh [11–13], accurate identification of a species macroscopically is difficult for ichthyologists as well, leading to errors in lists of species included in the literature. It is evident from several publications and review papers that, Bangladesh's taxonomy is out-of-date and not harmonized, when compared with that of the neighboring countries [14]. Hence, an integrative taxonomy method incorporating molecular and morphological data is required to overcome this obstacle and provide a seamless and widely recognized list of species inhabiting this nation. Researchers describe additional new species of fish each year, but it is not clear as to how many species are endemic to this nation.

In the current study, the authors have conducted phylogenetic analyses on fish specimens available in fish markets and natural resources nationwide to examine freshwater fish biodiversity in Bangladesh. Whenever necessary, morphological data has been used to identify cryptic or candidate species. The DNA-based barcoding method has proven to be an important molecular tool for non-specialists for species identification [15], and the mitochondrial cytochrome *c* subunit I (COI) sequence is a dependable barcode marker for measuring and assessing the taxonomic status of fish species [14,16–18]. This study attempts to review the biodiversity of Bangladesh's freshwater fishes, and explore the possible presence of cryptic species using the DNA barcoding technique.

## Materials and methods

### Study area, duration and identification of sample

A total of 124 fish specimens were collected from all regions of Bangladesh (see **Table 1**), photographed and their sex was recorded. All specimens were transferred at the "Evolution and Diversity Research Laboratory" at Jamalpur Science and Technology University, located in Jamalpur, Bangladesh. A voucher number was assigned to each specimen (please refer to **Table 1**). For subsequent molecular research, the species were maintained in either saturated DMSO/NaCl solution or 95% ethanol. Table 1 lists the study sites in detail. Initial identification of the collected fish species was based on current taxonomy knowledge and literature dealing with the fisheries science [4]. If necessary, the names of the species that have been accepted by the Catalog of Fishes online portal have been adhered to [6].

### DNA extraction and sequencing

Genomic DNA of all fish specimens was extracted and purified from conserved fin tissues using Chelex 100 [19]. Using the primers Fish-F1, Fish-F2, Fish-R1, and Fish-R2, the partial COI fragment was amplified by polymerase chain reaction (PCR) using EmeraldAmp PCR Master Mix (Takara Bio) [20]. Cycle conditions described in a previously conducted study [14] were followed for performing PCR. The amplified PCR products were purified using polyethylene glycol precipitation. Cycle sequencing procedures were then performed using the BigDye Terminator v3.1 Cycle Sequencing Kit (Applied Bioscience). Sequencing was performed using ABI 3500 automated sequencers. The newly discovered sequences were incorporated in the International Nucleotide Sequence Databases (INSD) through the DNA Data Bank of Japan (Accession numbers LC823243–LC823366).

**Table 1. Specimens used and identified COI haplotypes found in this study.**

| Voucher Number | Scientific Name | Family | Local Name | Location | Latitude and Longitude | Sample Collection Date | Accession number |
|---|---|---|---|---|---|---|---|
| MHBSFMSTU Fish 1 | *Cabdio morar* | Cyprinadae | Pieli | Sariakandi, Bogra | 24°55'08.4"N 89°38'26.0"E | 01-07-2018 | LC823243 |
| MHBSFMSTU Fish 2 | *Tenualosa ilisha* | Clupeidae | Ilish | Sariakandi, Bogra | 24°55'08.4"N 89°38'26.0"E | 01-07-2018 | LC823244 |
| MHBSFMSTU Fish 3 | *Ailia* sp. | Ailiidae | Bash patari/ Kajuli | Sariakandi, Bogra | 24°55'08.4"N 89°38'26.0"E | 01-07-2018 | LC823245 |
| MHBSFMSTU Fish 5 | *Pisodonophisboro* | Ophichthidae | Bamush/ Kharu | Sariakandi, Bogra | 24°55'08.4"N 89°38'26.0"E | 01-07-2018 | LC823246 |
| MHBSFMSTU Fish 6 | *Awaous* sp. | Gobiidae | Bochabele | Sariakandi, Bogra | 24°55'08.4"N 89°38'26.0"E | 01-07-2018 | LC823247 |
| MHBSFMSTU Fish 7 | *Mastacembelusar-matus* | Mastacembelidae | ShalBaim | Sariakandi, Bogra | 24°55'08.4"N 89°38'26.0"E | 01-07-2018 | LC823248 |
| MHBSFMSTU Fish 8 | *Botiadario* | Cobitidae | Boumach | Sariakandi, Bogra | 24°55'08.4"N 89°38'26.0"E | 01-07-2018 | LC823249 |
| MHBSFMSTU Fish 9 | *Channaorientalis* | Channidae | Raga | Jaliarhaor, Netrokona | 24°41'31.3"N 90°51'46.5"E | 25-08-2018 | LC823250 |
| MHBSFMSTU Fish 10 | *Aplocheiluspanchax* | Aplocheilidae | Kanpona | DangarBeel, Jamalpur | 24°55'06.5"N 89°51'03.6"E | 02-02-2019 | LC823251 |
| MHBSFMSTU Fish 11 | *Trichogaster lalius* | Osphronemidae | Khoilsha | DangarBeel, Jamalpur | 24°55'06.5"N 89°51'03.6"E | 02-02-2019 | LC823252 |
| MHBSFMSTU Fish 12 | *Trichogaster lalius* | Osphronemidae | Borokhoilsha | DangarBeel, Jamalpur | 24°55'06.5"N 89°51'03.6"E | 02-02-2019 | LC823253 |
| MHBSFMSTU Fish 13 | *Trichogaster chuna* | Osphronemidae | Lalkholisha | DangarBeel, Jamalpur | 24°55'06.5"N 89°51'03.6"E | 02-02-2019 | LC823254 |
| MHBSFMSTU Fish 14 | *Colisachuna* | Osphronemidae | ChunaKholisha | DangarBeel, Jamalpur | 24°55'06.5"N 89°51'03.6"E | 02-02-2019 | LC823255 |
| MHBSFMSTU Fish 15 | *Trichogaster chuna* | Osphronemidae | Lal Kholisa | DangarBeel, Jamalpur | 24°55'06.5"N 89°51'03.6"E | 02-02-2019 | LC823256 |
| MHBSFMSTU Fish 16 | *Macrognathusaral* | Mastacembelidae | Tara Baim | Jaliarhaor, Netrokona | 24°41'31.3"N 90°51'46.5"E | 25-08-2018 | LC823257 |
| MHBSFMSTU Fish 17 | *Lepidocephal-ichthyssp.* | Cobitidae | Morichapuiya/ Gutum | Jaliarhaor, Netrokona | 24°41'31.3"N 90°51'46.5"E | 25-08-2018 | LC823258 |
| MHBSFMSTU Fish 18 | *Ompokpabda* | Siluridae | Pabda | Gaglajur bazar, Netrokona | 24°52'36.2"N 91°31'38.4"E | 25-08-2018 | LC823259 |
| MHBSFMSTU Fish 19 | *Mystus cavasius* | Bagridae | Gulsha | Jaliarhaor, Netrokona | 24°41'31.3"N 90°51'46.5"E | 25-08-2018 | LC823260 |
| MHBSFMSTU Fish 20 | *Mystus tengara* | Bagridae | Tengra | Jaliarhaor, Netrokona | 24°41'31.3"N 90°51'46.5"E | 25-08-2018 | LC823261 |
| MHBSFMSTU Fish 22 | *Mystus carcio* | Bagridae | Bujuri | Jaliarhaor, Netrokona | 24°41'31.3"N 90°51'46.5"E | 25-08-2018 | LC823262 |
| MHBSFMSTU Fish 23 | *Channamarulius* | Channidae | Gojar | Ghaglajur, Netrokona | 24°52'36.2"N 91°31'38.4"E | 15-12-2018 | LC823263 |
| MHBSFMSTU Fish 25 | *Nandus nandus* | Nandidae | Bheda | Jaliarhaor, Netrokona | 24°41'31.3"N 90°51'46.5"E | 25-08-2018 | LC823264 |
| MHBSFMSTU Fish 26 | *Channapunctata* | Channidae | Taki | Jaliarhaor, Netrokona | 24°41'31.3"N 90°51'46.5"E | 25-08-2018 | LC823265 |
| MHBSFMSTU Fish 27 | *Mystus cavasius* | Bagridae | Kabashi Tengra | Mawaghat Munshigonj | 23°27'59.3"N 90°17'22.1"E | 25-08-2018 | LC823266 |
| MHBSFMSTU Fish 28 | *Bagariusbagarius* | Sisoridae | Bagha air | Kongso, Netrokona | 24°52'50.2"N 90°44'03.2"E | 25-08-2018 | LC823267 |
| MHBSFMSTU Fish 30 | *Sperata aorella* | Bagridae | Air | Kongso, Netrokona | 24°52'50.2"N 90°44'03.2"E | 25-08-2018 | LC823268 |

*(Continued)*

**Table 1.** (Continued)

| Voucher Number | Scientific Name | Family | Local Name | Location | Latitude and Longitude | Sample Collection Date | Accession number |
|---|---|---|---|---|---|---|---|
| MHBSFMSTU Fish 31 | *Trichopsisvittata* | Osphronemidae | PokaKholisha | MawaghatMunshigonj | 23°27'59.3"N 90°17'22.1"E | 25-08-2018 | LC823269 |
| MHBSFMSTU Fish 32 | *Parambassislala* | Ambassidae | Lalchanda | Jaliarhaor, Netrokona | 24°41'31.3"N 90°51'46.5"E | 25-08-2018 | LC823270 |
| MHBSFMSTU Fish 34 | *Xenentodoncancila* | Belonidae | Kakila | Jaliarhaor, Netrokona | 24°41'31.3"N 90°51'46.5"E | 25-08-2018 | LC823271 |
| MHBSFMSTU Fish 36 | *Macrognathus pancalus* | Mastacembelidae | Chikra/ Gujibaim | Jaliarhaor, Netrokona | 24°41'31.3"N 90°51'46.5"E | 25-08-2018 | LC823272 |
| MHBSFMSTU Fish 37 | *Badis badis* | Badidae | Napit koi | Jaliarhaor, Netrokona | 24°41'31.3"N 90°51'46.5"E | 25-08-2018 | LC823273 |
| MHBSFMSTU Fish 38 | *Chacachaca* | Chacidae | Bengachaca | Gaglajur Bazar, Netrokona | 24°52'36.2"N 91°31'38.4"E | 25-08-2018 | LC823274 |
| MHBSFMSTU Fish 39 | *Notopterusnotopterus* | Notopteridae | Foli | Balikhola, Kishorganj | 24°13'27.8"N 91°06'32.2"E | 25-08-2018 | LC823275 |
| MHBSFMSTU Fish 40 | *Labeorohita* | Cyprinidae | Rui | Sokal bazar, Jamalpur | 24°55'47.4"N 89°56'45.2"E | 17-04-2019 | LC823276 |
| MHBSFMSTU Fish 41 | *Labeocalbasu* | Cyprinidae | Kali baush | Sokal bazar, Jamalpur | 24°55'47.4"N 89°56'45.2"E | 17-04-2019 | LC823277 |
| MHBSFMSTU Fish 46 | *Esomusdanrica* | Cyprinidae | Darkina | Sokal bazar, Jamalpur | 24°55'47.4"N 89°56'45.2"E | 25-08-2018 | LC823278 |
| MHBSFMSTU Fish 47 | *Setipinnaphasa* | Engraulidae | Chibuk chela/ Phasha | Guthail, Jamalpur | 25°05'13.3"N 89°42'49.0"E | 25-08-2018 | LC823279 |
| MHBSFMSTU Fish 48 | *Catlacatla* | Cyprinidae | Katol | Sokal bazar, Jamalpur | 24°55'47.4"N 89°56'45.2"E | 17-04-2019 | LC823280 |
| MHBSFMSTU Fish 49 | *Cirrhinusmrigala* | Cyprinidae | Mrigal | Sanondobari, Jamalpur | 25°22'51.3"N 89°44'40.8"E | 17-04-2019 | LC823281 |
| MHBSFMSTU Fish 50 | *Daniorerio* | Cyprinidae | Zebra anju/ Zebra | Kongsho, Sherpur | 24°59'52.2"N 90°00'29.6"E | 17-04-2019 | LC823282 |
| MHBSFMSTU Fish 51 | *Coricasoborna* | Clupeidae | Subornakaski | Kongso, Sherpur | 24°59'52.2"N 90°00'29.6"E | 15-03-2019 | LC823283 |
| MHBSFMSTU Fish 52 | *Salmostoma phulo* | Cyprinidae | Narikeli Chela/ Katari | Sokal bazar, Jamalpur | 24°55'47.4"N 89°56'45.2"E | 02-02-2019 | LC823284 |
| MHBSFMSTU Fish 53 | *Hyporhamphus limbatus* | Hemiramphidae | Ekthota | Kendua, Netrokona | 24°39'49.9"N 90°50'28.7"E | 15-03-2019 | LC823285 |
| MHBSFMSTU Fish 54 | *Channastriata* | Channidae | Shol | Kendua, Netrokona | 24°39'49.9"N 90°50'28.7"E | 02-02-2019 | LC823286 |
| MHBSFMSTU Fish 55 | *Pseudeutropiusatherinoides* | Schilbeidae | Batai | Kendua, Netrokona | 24°39'49.9"N 90°50'28.7"E | 22-04-2019 | LC823287 |
| MHBSFMSTU Fish 56 | *Ophichthys* sp. | Synbranchidae | kuchia | Sokal bazar, Jamalpur | 24°55'47.4"N 89°56'45.2"E | 02-02-2019 | LC823288 |
| MHBSFMSTU Fish 57 | *Wallagoattu* | Siluridae | Boal | Kongso, Netrokona | 24°52'50.2"N 90°44'03.2"E | 25-08-2018 | LC823289 |
| MHBSFMSTU Fish 59 | *Heteropneustesfossilis* | Heteropneustidae | Shing | Dangarbeel, Jamalpur | 24°55'06.5"N 89°51'03.6"E | 02-02-2019 | LC823290 |
| MHBSFMSTU Fish 61 | *Puntius sophore* | Cyprinidae | Jatpunti/ Bhadiputi | JaliarHaor, Netrokona | 24°41'31.3"N 90°51'46.5"E | 25-08-2018 | LC823291 |
| MHBSFMSTU Fish 63 | *Pethia conchonius* | Cyprinidae | Tit punti | Sokal bazar, Jamalpur | 24°55'47.4"N 89°56'45.2"E | 17-04-2019 | LC823292 |
| MHBSFMSTU Fish 64 | *Chandanama* | Ambassidae | Chanda (lomba) | Balikhula, Kishorgonj | 24°13'27.8"N 91°06'32.2"E | 17-04-2019 | LC823293 |

*(Continued)*

**Table 1.** (Continued)

| Voucher Number | Scientific Name | Family | Local Name | Location | Latitude and Longitude | Sample Collection Date | Accession number |
|---|---|---|---|---|---|---|---|
| MHBSFMSTU Fish 65 | *Parambassis ranga* | Ambassidae | Chanda (Gol)/ Phopa | Kendua, Netrokona | 24°39'49.9"N 90°50'28.7"E | 17-04-2019 | LC823294 |
| MHBSFMSTU Fish 66 | *Macrognathus pancalus* | Mastacembelidae | Guchibaim | Mawaghat, Munshigonj | 23°27'59.3"N 90°17'22.1"E | 17-04-2019 | LC823295 |
| MHBSFMSTU Fish 67 | *Amblypharyngodon mola* | Cyprinidae | Fakasemola | JaliarHaor, Netrokona | 24°41'31.3"N 90°51'46.5"E | 17-04-2019 | LC823296 |
| MHBSFMSTU Fish 69 | *Channagachua* | Channidae | Cheng | Dangarbeel, Jamalpur | 24°55'06.5"N 89°51'03.6"E | 17-04-2019 | LC823297 |
| MHBSFMSTU Fish 70 | *Salmostoma bacalia* | Cyprinidae | Ful chela | Sokal bazar, Jamalpur | 24°55'47.4"N 89°56'45.2"E | 17-04-2019 | LC823298 |
| MHBSFMSTU Fish 71 | *Eutropiichthysvacha* | Ailiidae | Bacha | Ghaglajur, Netrokona | 24°52'36.2"N 91°31'38.4"E | 25-08-2018 | LC823299 |
| MHBSFMSTU Fish 72 | *Gudusia chapra* | Clupeidae | GoniChapila | Sariakandi, Bogra | 24°55'08.4"N 89°38'26.0"E | 01-07-2018 | LC823300 |
| MHBSFMSTU Fish 73 | *Clupisomagarua* | Ailiidae | Ghaura | Mawaghat, Munshigonj | 23°27'59.3"N 90°17'22.1"E | 17-04-2019 | LC823301 |
| MHBSFMSTU Fish 74 | *Rita rita* | Bagridae | Rita | Balikhola, Kishorgonj | 24°13'27.8"N 91°06'32.2"E | 01-07-2018 | LC823302 |
| MHBSFMSTU Fish 75 | *Hemibagrusmenoda* | Bagridae | Hugli/ Arwari/ Ghagla | Ghaglajur bazaar, Netrokona | 24°52'36.2"N 91°31'38.4"E | 01-07-2018 | LC823303 |
| MHBSFMSTU Fish 77 | *Leiodon cutcutia* | Tetraodontidae | Patipotka | Bramaputra river, Jamalpur | 24°55'37.0"N 89°57'42.6"E | 10-11-2018 | LC823304 |
| MHBSFMSTU Fish 78 | *Devariodevario* | Cyprinidae | Hingra kata punti/ Debari/ Chapchela | Ghaglajur Bazar, Netrokona | 24°52'36.2"N 91°31'38.4"E | 02-02-2019 | LC823305 |
| MHBSFMSTU Fish 79 | *Osteobramacotio* | Cyprinidae | Dhela | Gaglajur Bazar, Netrokona | 24°52'36.2"N 91°31'38.4"E | 02-02-2019 | LC823306 |
| MHBSFMSTU Fish 80 | *Mylopharyngodon piceus* | Cyprinidae | Mohashol | Sokal Bazar, Jamalpur | 24°55'47.4"N 89°56'45.2"E | 25-08-2019 | LC823307 |
| MHBSFMSTU Fish 81 | *Labeo gonius* | Cyprinidae | Gonia | Sokal Bazar, Jamalpur | 24°55'47.4"N 89°56'45.2"E | 25-08-2019 | LC823308 |
| MHBSFMSTU Fish 82 | *Labeo gonius* | Cyprinidae | Tatkini/ Bhagna/ Raik/ Lacho/bata | Balikhula, Kishorgonj | 24°13'27.8"N 91°06'32.2"E | 25-08-2019 | LC823309 |
| MHBSFMSTU Fish 83 | *Erethistes* sp. | Sisoridae | Kutakanti | Ghaglajur Bazar, Netrokona | 24°52'36.2"N 91°31'38.4"E | 25-08-2019 | LC823310 |
| MHBSFMSTU Fish 84 | *Gagatayoussoufi* | Sisoridae | Gum/ Gagata | Ghaglajur, Netrokona | 24°52'36.2"N 91°31'38.4"E | 25-08-2019 | LC823311 |
| MHBSFMSTU Fish 85 | *Pethia gelius* | Cyprinidae | Gilipunti/ puti | Ghaglajur Bazar, Netrokona | 24°52'36.2"N 91°31'38.4"E | 07-01-2020 | LC823312 |
| MHBSFMSTU Fish 86 | *Rhinomugil corsula* | Mugilidae | Folla, khalla, halla, khorsula | Balikhula, Kishorgonj | 24°13'27.8"N 91°06'32.2"E | 01-01-2020 | LC823313 |
| MHBSFMSTU Fish 87 | *Chandramara chandramara* | Bagridae | Jolbujuri | Ghaglajur Bazar, Netrokona | 24°52'36.2"N 91°31'38.4"E | 07-01-2020 | LC823314 |
| MHBSFMSTU Fish 90 | *Silonia silondia* | Ailiidae | Shilong | Kawniar char, Dewangonj | 25°23'59.3"N 89°47'49.0"E | 15-01-2020 | LC823315 |
| MHBSFMSTU Fish 91 | *Glyptothorax telchitta* | Sisoridae | Lalkutakanti | Kawniar char, Dewangonj | 25°23'59.3"N 89°47'49.0"E | 15-01-2020 | LC823316 |
| MHBSFMSTU Fish 92 | *Gagatacenia* | Sisoridae | Kutakanti, Cenia, Jangla, Kaowa, Gang tengra | Sanondabar, Jamalpur | 25°22'51.3"N 89°44'40.8"E | 25-01-2020 | LC823317 |
| MHBSFMSTU Fish 93 | *Gogangra laevis* | Sisoridae | Gang Tengra | Dewangonj bazar, Jamalpur | 25°10'01.7"N 89°45'57.5"E | 15-01-2020 | LC823318 |

*(Continued)*

**Table 1.** (Continued)

| Voucher Number | Scientific Name | Family | Local Name | Location | Latitude and Longitude | Sample Collection Date | Accession number |
|---|---|---|---|---|---|---|---|
| MHBSFMSTU Fish 96 | *Rasboradaniconius* | Cyprinidae | Darkina | Kendua, Netrokna | 24°39'49.9"N 90°50'28.7"E | 22-01-2020 | LC823319 |
| MHBSFMSTU Fish 97 | *Botialohachata* | Cobitidae | Bou/Rani | Sokal bazar, Jamalpur | 24°55'47.4"N 89°56'45.2"E | 22-01-2020 | LC823320 |
| MHBSFMSTU Fish 98 | *Labeo boggut* | Cyprinidae | Vangon/ Gonari | Sokal bazar, Jamalpur | 24°55'47.4"N 89°56'45.2"E | 22-01-2020 | LC823321 |
| MHBSFMSTU Fish 99 | *Paracanthocobitis abutwebi* | Nemacheilidae | Balichata/ Puiya | Sanondabari, Jamalpur | 25°22'51.3"N 89°44'40.8"E | 25-01-2020 | LC823322 |
| MHBSFMSTU Fish 100 | *Schistura* sp. | Nemacheilidae | Balikhura/ Puiya | Sanondabari, Jamalpur | 25°22'51.3"N 89°44'40.8"E | 25-01-2020 | LC823323 |
| MHBSFMSTU Fish 101 | *Anodontostomacha-cunda* | Clupeidae | Chacunda | Defla Bazar, Jamalpur | 25°01'17.1"N 89°50'16.3"E | 25-01-2020 | LC823324 |
| MHBSFMSTU Fish 102 | *Salmostoma bacalia* | Cyprinidae | Chela | Soal bazar, Jamalpur | 24°55'47.4"N 89°56'45.2"E | 21-12-2020 | LC823325 |
| MHBSFMSTU Fish 103 | *Badis badis* | Badidae | Napit koi | Sokal bazar, Jamalpur | 24°55'47.4"N 89°56'45.2"E | 01-02-2020 | LC823326 |
| MHBSFMSTU Fish 104 | *Rhinomugil corsula* | Mugilidae | Flathead mullet | Sokal Bazar, Jamalpur | 24°55'47.4"N 89°56'45.2"E | 01-02-2020 | LC823327 |
| MHBSFMSTU Fish 105 | *Pethia conchonius* | Cyprinidae | Kanchonputi/ Taka puti | Guthail, Jamalpur | 25°05'13.3"N 89°42'49.0"E | 05-02-2020 | LC823328 |
| MHBSFMSTU Fish 106 | *Neoeucirrhichthys maydelli* | Cobitidae | GualParapuiya | Guthail, Jamalpur | 25°05'13.3"N 89°42'49.0"E | 05-02-2020 | LC823329 |
| MHBSFMSTU Fish 107 | *Chela laubuca* | Cyprinidae | Chep chela | Guthail, Jamalpur | 25°05'13.3"N 89°42'49.0"E | 05-02-2020 | LC823330 |
| MHBSFMSTU Fish 108 | *Tariqilabeo latius* | Cyprinidae | Matikhora/kalabata | Guthail, Jamalpur | 25°05'13.3"N 89°42'49.0"E | 05-02-2020 | LC823331 |
| MHBSFMSTU Fish 109 | *Pangio Pangia* | Cobitidae | Kuttapuiya/Panga/ Kolipuiya | Guthail, Jamalpur | 25°05'13.3"N 89°42'49.0"E | 05-02-2020 | LC823332 |
| MHBSFMSTU Fish 110 | *Odontamblyopus rubicundus* | Gobiidae | Lalchewa | Mawaghat, Munshigonj | 23°27'59.3"N 90°17'22.1"E | 08-01-2020 | LC823333 |
| MHBSFMSTU Fish 111 | *Apocryptes bato* | Gobiidae | Chewa/ Chiring | Mawaghat, Munshigonj | 23°27'59.3"N 90°17'22.1"E | 08-01-2020 | LC823334 |
| MHBSFMSTU Fish 112 | *Sperata aor* | Bagridae | Guizza air | Mawaghat, Munshigonj | 23°27'59.3"N 90°17'22.1"E | 08-01-2020 | LC823335 |
| MHBSFMSTU Fish 118 | *Ompok bimaculatus* | Siluridae | Kala pabda | Rangamati | 22°58'00.8"N 92°12'05.6"E | 11-01-2020 | LC823336 |
| MHBSFMSTU Fish 121 | *Latescalcarifer* | Centropomidae | Deshivetki | PaikariMotshoArot, Barishal | 22°42'13.4"N 90°22'32.4"E | 27-01-2021 | LC823337 |
| MHBSFMSTU Fish 126 | *Cynoglossus* sp. | Cynoglossidae | Kukurjib | Fishery Ghat, Chittagong | 22°19'43.9"N 91°50'49.6"E | 11-02-2021 | LC823338 |
| MHBSFMSTU Fish 130 | *Polynemusparadiseus* | Polynemidae | Taposhi | PaikariMotshoArot, Barishal | 22°42'13.4"N 90°22'32.4"E | 27-01-2021 | LC823339 |
| MHBSFMSTU Fish 134 | *Tariqilabeo latius* | Cyprinidae | Kalobata | Shampurnagarghat, Rajshahi | 24°20'55.7"N 88°39'26.5"E | 22-12-2020 | LC823340 |
| MHBSFMSTU Fish 135 | *Psilorhynchus sucatio* | Psilorhynchidae | Titari | CharlokkhipurShibgonj, Chapainawabgonj | 24°35'56.2"N 88°05'20.1"E | 21-12-2020 | LC823341 |
| MHBSFMSTU Fish 136 | *Glossogobius giuris* | Gobiidae | Bele | Narayanpur, Shibgonj, Chapainawabgonj | 24°34'05.7"N 88°05'48.8"E | 21-12-2020 | LC823342 |

*(Continued)*

**Table 1.** (Continued)

| Voucher Number | Scientific Name | Family | Local Name | Location | Latitude and Longitude | Sample Collection Date | Accession number |
|---|---|---|---|---|---|---|---|
| MHBSFMSTU Fish 137 | *Setipinna phasa* | Engraulidae | Pasha | Charlokkhipur, Shibgonj, Chapai Nawabgonj | 24°35'56.2"N 88°05'20.1"E | 21-12-2020 | LC823343 |
| MHBSFMSTU Fish 138 | *Terapon jarbua* | Terapontidae | Rekha | Shondhya bazar, Moylaputa,Khulna | 22°48'46.7"N 89°33'25.6"E | 28-01-2021 | LC823344 |
| MHBSFMSTU Fish 139 | *Cynoglossus* sp. | Cynoglossidae | Kukurjib | RupshaGhat, Khulna | 22°48'05.5"N 89°34'52.1"E | 28-01-2021 | LC823345 |
| MHBSFMSTU Fish 140 | *Maculabatis pastinacoides* | Dasyatidae | Shaplapata | PaikarimothsoArot, Barishal | 22°42'13.4"N 90°22'32.4"E | 27-01-2021 | LC823346 |
| MHBSFMSTU Fish 141 | *Stigmatogobius sadanandio* | Gobiidae | Nondibaila | RupshaGhat, Khulna | 22°48'26.0"N 89°34'49.5"E | 28-01-2021 | LC823347 |
| MHBSFMSTU Fish 142 | *Acentrogobius viridipunctatus* | Gobiidae | Shobujfutkibaila | Sondha bazar, Rupsha, Khulna | 22°48'47.1"N 89°33'25.0"E | 28-01-2021 | LC823348 |
| MHBSFMSTU Fish 143 | *Apocryptes bato* | Gobiidae | Chiring | Sondha bazar, Rupsha, Khulna | 22°48'47.1"N 89°33'25.0"E | 28-01-2021 | LC823349 |
| MHBSFMSTU Fish 145 | *Trypauchen vagina* | Gobiidae | Raja chewa | RupshaGhat, Khulna | 22°48'05.5"N 89°34'52.1"E | 28-01-2021 | LC823350 |
| MHBSFMSTU Fish 146 | *Platycephalusindicus* | Platycephalidae | Mur baila | RupshaGhat, Khulna | 22°48'05.5"N 89°34'52.1"E | 28-01-2021 | LC823351 |
| MHBSFMSTU Fish 147 | *Eleutheronematetrad-actylum* | Polynemidae | Tailla | RupshaGhat, Khulna | 22°48'05.5"N 89°34'52.1"E | 28-01-2021 | LC823352 |
| MHBSFMSTU Fish 148 | *Mystusgulio* | Bagridae | Nunatengra | PaikarimothsoArot,Barishal | 22°42'13.4"N 90°22'32.4"E | 27-01-2021 | LC823353 |
| MHBSFMSTU Fish 149 | *Sillaginopsis domina* | Sillaginidae | Kulardati/ Tulardanti | PaikarimothsoArot, Barishal | 22°42'13.4"N 90°22'32.4"E | 27-01-2021 | LC823354 |
| MHBSFMSTU Fish 150 | *Toxoteschatareus* | Toxotidae | Deha | RupshaGhat, Khulna | 22°48'05.5"N 89°34'52.1"E | 25-01-2021 | LC823355 |
| MHBSFMSTU Fish 151 | *Arius gagora* | Ariidae | Jonglitengra | RupshaGhat, Khulna | 22°48'05.5"N 89°34'52.1"E | 28-01-2021 | LC823356 |
| MHF154 | *Puntius sophore* | Cyprinidae | Chela puti | HaziShariotullah Market, Faridpur | 23°36'04.4"N 89°49'51.1"E | 26-01-2021 | LC823357 |
| MHBSFMSTU Fish 156 | *Gagatagagata* | Sisoridae | Gang tengra/ Deshitengra | PaikariMotshoArot, Barishal | 22°42'13.4"N 90°22'32.4"E | 27-01-2021 | LC823358 |
| MHBSFMSTU Fish 157 | *Awaous* sp. | Gobiidae | Chewa/Bele | Tepakhula Bazar, Faidpur | 23°36'48.0"N 89°51'22.8"E | 26-01-2021 | LC823359 |
| MHBSFMSTU Fish 158 | *Glossogobius giuris* | Gobiidae | Lalbele | Tepakhula Bazar, Faidpur | 23°36'48.0"N 89°51'22.8"E | 26-01-2021 | LC823360 |
| MHBSFMSTU Fish 159 | *Glossogobius giuris* | Gobiidae | Bele | Tepakhula Bazar, Faidpur | 23°36'48.0"N 89°51'22.8"E | 26-01-2021 | LC823361 |
| MHBSFMSTU Fish 161 | *Acanthopagruslatus* | Sparidae | Datina | Chakaria, Chittagong | 21°45'04.2"N 92°01'11.0"E | 26-01-2021 | LC823362 |
| MHBSFMSTU Fish 163 | *Coiliaramcarati* | Engraulidae | Holudolua | Chakaria, Chittagong | 21°45'04.2"N 92°01'11.0"E | 27-01-2021 | LC823363 |
| MHBSFMSTU Fish 164 | *Nandus nandus* | Nandidae | Meni | Ilisharot, Rangamati | 22°38'54.8"N 92°11'05.0"E | 30-01-2021 | LC823364 |
| MHBSFMSTU Fish 301 | *Amblypharyngodon mola* | Cyprinidae | Lalmola | Mawaghat, Munshigonj | 23°27'59.3"N 90°17'22.1"E | 08-01-2020 | LC823365 |
| MHBSFMSTU Fish 302 | *Awaous* sp. | Gobiidae | Bele Mach | Tepakhula Bazar, Faidpur | 23°36'48.0"N 89°51'22.8"E | 26-01-2021 | LC823366 |

## Phylogenetical analysis

The resulting COI gene nucleotide sequences were matched with additional COI sequences of Bangladeshi fish from INSD [14] and the MAFFT (Multiple Alignment using Fast Fourier Transform) v. 7.427 with the L-INS-I option [21]. Phylogenetic trees were inferred using maximum likelihood (ML) and Bayesian inference (BI) techniques. IQ-TREE v. 1.6.12 was used to estimate nucleotide substitution models for ML and BI analyses using AICc [22]. The IQ-TREE was used to determine the ML phylogeny, and 1000 pseudo replicates were used for an ultra-fast bootstrapping (BS). MrBayes v. 3.2.6 was used to obtain the Bayesian posterior probabilities and BI tree [23]. A tree sample was obtained after every 1000 generations, throughout two separate runs of four Markov chains totaling 100 million generations. Tracer v. 1.7.1 [24] was used to examine the parameter estimates and convergence, and the first 10% trees were eliminated in light of the findings. MEGA11'spairwise-deletion option was used for computing pairwise comparisons of the uncorrected p-distance [25] (S1 Table).

## Species delimitation analyses and morphology measurement

Species delimitation investigations were conducted using Assemble Species by Automatic Partitioning (ASAP) [26] based on 712 COI sequences used in phylogenetic analysis (S2 Table). The ASAP analysis was conducted using the online ASAP version (https://bioinfo.mnhn.fr/abi/public/asap/asapweb.html) using the uncorrected pairwise distances with the default parameters. We looked at morphological parameters under both a microscope and a magnifying glass. We only took morphometric measurements of the chosen cryptic fish using a digital vernier caliper (Mitutoyo, accuracy ±0.02 mm).

## Ethics statement

The specimens were obtained from either from markets or fishermen, or they were collected from the wild using a beach seine or hand net. The fish were then euthanized by immersion in buffered tricaine-methanesulphonate (MS 222) for thirty minutes after the animals stopped moving, in accordance with the guidelines outlined in the permits issued by Jamalpur Science and Technology University's Ethics Committee (Letter no. 37.01.0044.064.05.001.24.553).

# Results

Fish specimens were gathered nationwide from more than 28 locations (Table 1), and analyzed the mitochondrial COI sequences of 124 individuals representing 121 species, 94 genera, 39 families, and 19 orders. Sequences with ≥1 nucleotide change were considered distinct haplotypes. Our COI alignment matrix consisted of 655 bp representing 119 haplotypes. Table 1 provides a summary of all haplotypes. Based on COI sequences, nearly identical topologies were observed in the ML (ln $L = -59788.54$) and BI (ln $L = -60808.92$) trees (S1 Fig).

Uncorrected p-distance and species delimitation analysis demonstrated 243 groups with the best score. Most of the OTUs (operational taxonomy units) sequenced in this study were assigned to groups with known species; however, some specimens formed their own unique groups. In particular, seven OTUs had to be assigned to different groups, despite the presence of the same species identified morphologically, indicating that these specimens were cryptic species (Table 2).

Specifically, the specimen of *Ailia* sp. (Ailiidae) had been morphologically identified as *Ailia coila*, but genetically it was found to belong to a different species (*Ailia* sp.) compared to its nearest congener (COI divergence 8.2%) (Fig 1). This specimen was collected from the area adjacent to the Jamuna River (Sariakandi, Bogura). Our examined specimens of genus *Awaous*

**Table 2. Possible cryptic species found in this study.**

| Voucher Number | Scientific Name (Family) | Morphological Characteristics | Total length (mm) | Ratios | Accession Number |
|---|---|---|---|---|---|
| MHBSFMSTU Fish 3 | *Ailia* sp. (Ailiidae) | 1. Body elongate, deeply compressed and silvery in color<br>2. Four pairs of barbells present.<br>3. Adipose fin present.<br>4. Lateral line extends up to tip of the caudal fin. | 78 | TL:SL=7.8:7.2<br>TL:HB=7.8:1.3<br>TL:LBD=7.8:0.4<br>TL:HL=7.8:1.1<br>TL:ED=7.8:0.3<br>TL:PROL=7.8:0.2<br>TL:PTOL=7.8:0.6 | LC823245 |
| MHBSFMSTU Fish 6 | *Awaous* sp. (Gobiidae) | 1. Elongate, sub-cylindrical anteriorly, compressed posteriorly<br>2. Head obtusely convex and mouth little oblique<br>3. Spiny dorsal fin.<br>4. Opercule scaled. | 99 | TL:SL=9.9:8.2<br>TL:HB=9.9:1.9<br>TL:LBD=9.9:0.8<br>TL:HL=9.9:2.1<br>TL:ED=9.9:0.4<br>TL:PROL=9.9:0.7<br>TL:PTOL=9.9:1.0 | LC823247 |
| MHBSFMSTU Fish 17 | *Lepidocephal-ichthys*sp. (Cobitidae) | 1. Caudal fin rounded; a light band extends from snout to caudal<br>2. Deep black spots present pectoral region and these spots become irregular at the base of caudal peduncle.<br>3. Small pelvic fin present. | 69 | TL:HB=6.9:1.2<br>TL:LBD=6.9:0.8<br>TL:HL=6.9:1.1<br>TL:ED=6.9:0.2<br>TL:PROL=6.9:0.4<br>TL:PTOL=6.9:1.2 | LC823258 |
| MHBSFMSTU Fish 56 | *Ophichthys* sp. (Synbranchidae) | 1. Body cylindrical, elongated with rounded abdomen.<br>2. Small eye and head not conspicuous.<br>3. Tail tapering and compressed.<br>4. Upper jaw longer, fleshy lips. | 256 | TL:LBD=25.5:1.3<br>TL:HL=25.5:0.5<br>TL:ED=25.5:0.1<br>TL:PROL=25.:0.3 | LC823288 |
| MHBSFMSTU Fish 83 | *Erethistes* sp. (Sisoridae) | 1. Head depressed and flattened ventrally.<br>2. Dorsal profile arched and ventral profile nearly horizontal.<br>3. Mouth small inferior and upper jaw little longer.<br>4. Four pairs of barbells and pectoral girdle present | 57 | TL:SL=5.7:4.7<br>TL:HB=5.7:1.1<br>TL:LBD=5.7:0.3<br>TL:HL=5.7:1.5<br>TL:ED=5.7:0.1<br>TL:PROL=5.7:0.8<br>TL:PTOL=5.7:0.6 | LC823309 |
| MHBSFMSTU Fish 100 | *Schistura*sp. (Nemacheilidae) | 1. Snout conical and abdomen rounded.<br>2. Eyes dorso-lateral.<br>3. Nostril in front of eye separated by valve.<br>4. Caudal fin weakly forked. | 38 | TL:SL=3.8:3.3<br>TL:HB=3.8:0.5<br>TL:LBD=3.8:0.1<br>TL:HL=3.8:0.6<br>TL:ED3.8:0.1<br>TL:PROL=3.8:0.2<br>TL:PTOL=3.8:0.3 | LC823323 |
| MHBSFMSTU Fish 126 | *Cynoglossus* sp. (Cynoglossidae) | 1. Body elongated, flat and tapering posterioly.<br>2. Eyes close together, on the left side of the body, upper one is advanced compared to lower one<br>3. Snout obtusely pointed.<br>4. Dorsal and anal fins confluent with caudal. | 291 | TL:SL=29.1:27.2<br>TL:HB=29.1:6.1<br>TL:LBD=29.1:0.2<br>TL:HL=3.8:4.8<br>TL:ED=29.1:0.5<br>TL:PROL=29.1:0.9<br>TL:PTOL=29.1:3.4 | LC823337 |

formed clades with other *Awaous* sp. (Gobiidae) and demonstrated poor degrees of divergence (COI divergence 0.3%) (Fig 2). These specimens were collected from the Bogura and Faridpur districts which are linked with the rivers Jamuna and Padma. *Lepidocephalichthys* sp. (Cobitidae) collected from Netrokona (24°41'31.3"N 90°51'46.5"E), formed clades with other known sequences (MK572291, MK572294) and demonstrated15.50% COI divergence from its nearest congeners (Fig 1). *Ophichthys* sp.(Synbranchidae) demonstrated 7.7% genetic divergence from its near congeners (Fig 1). *Erethistes* sp. (Sisoridae), belonging to the genus

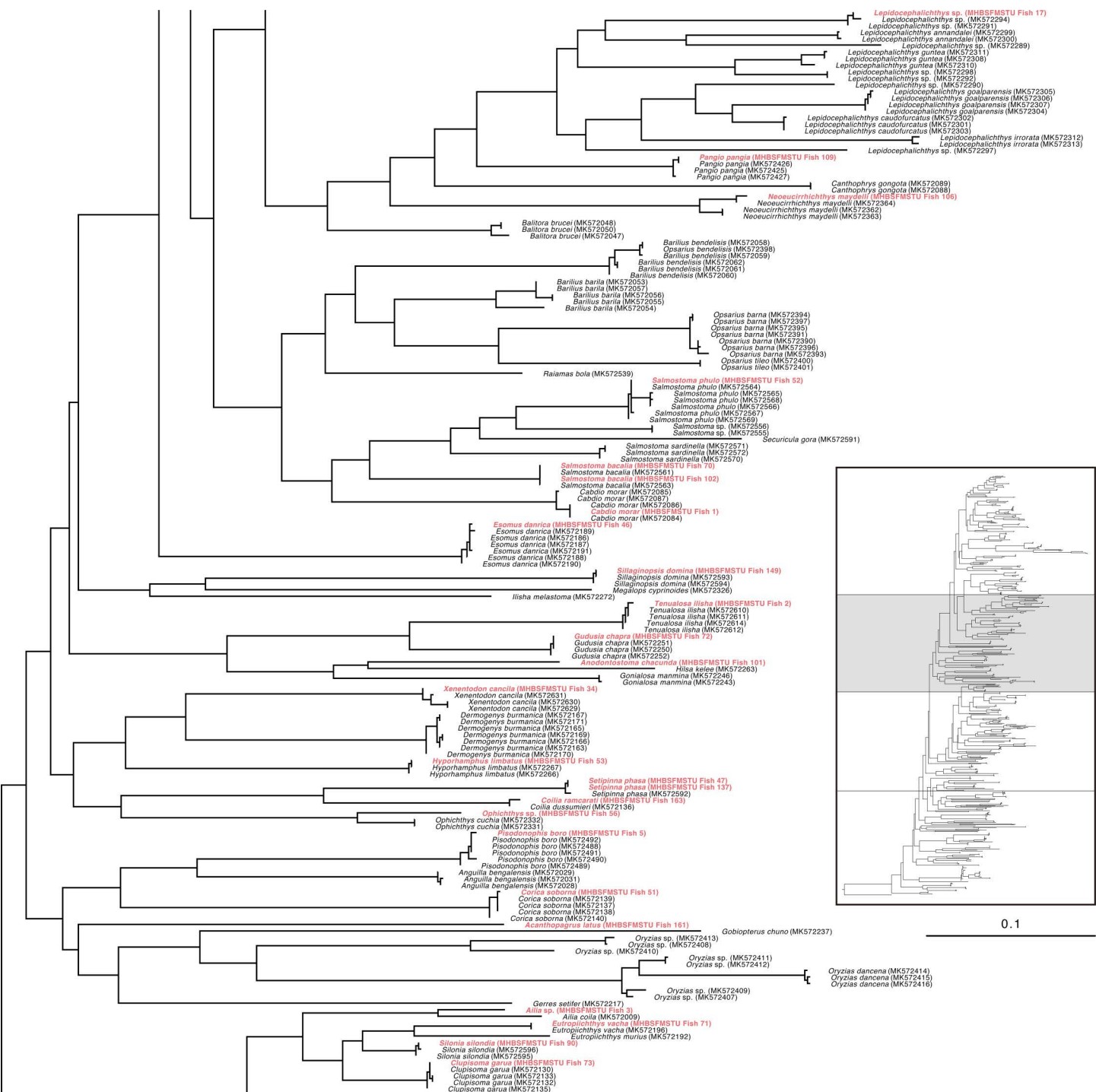

**Fig 1. Maximum Likelihood (ML) tree of freshwater species of Bangladesh based on nucleotide sequences of COI gene (partial portion of S1 Fig).**

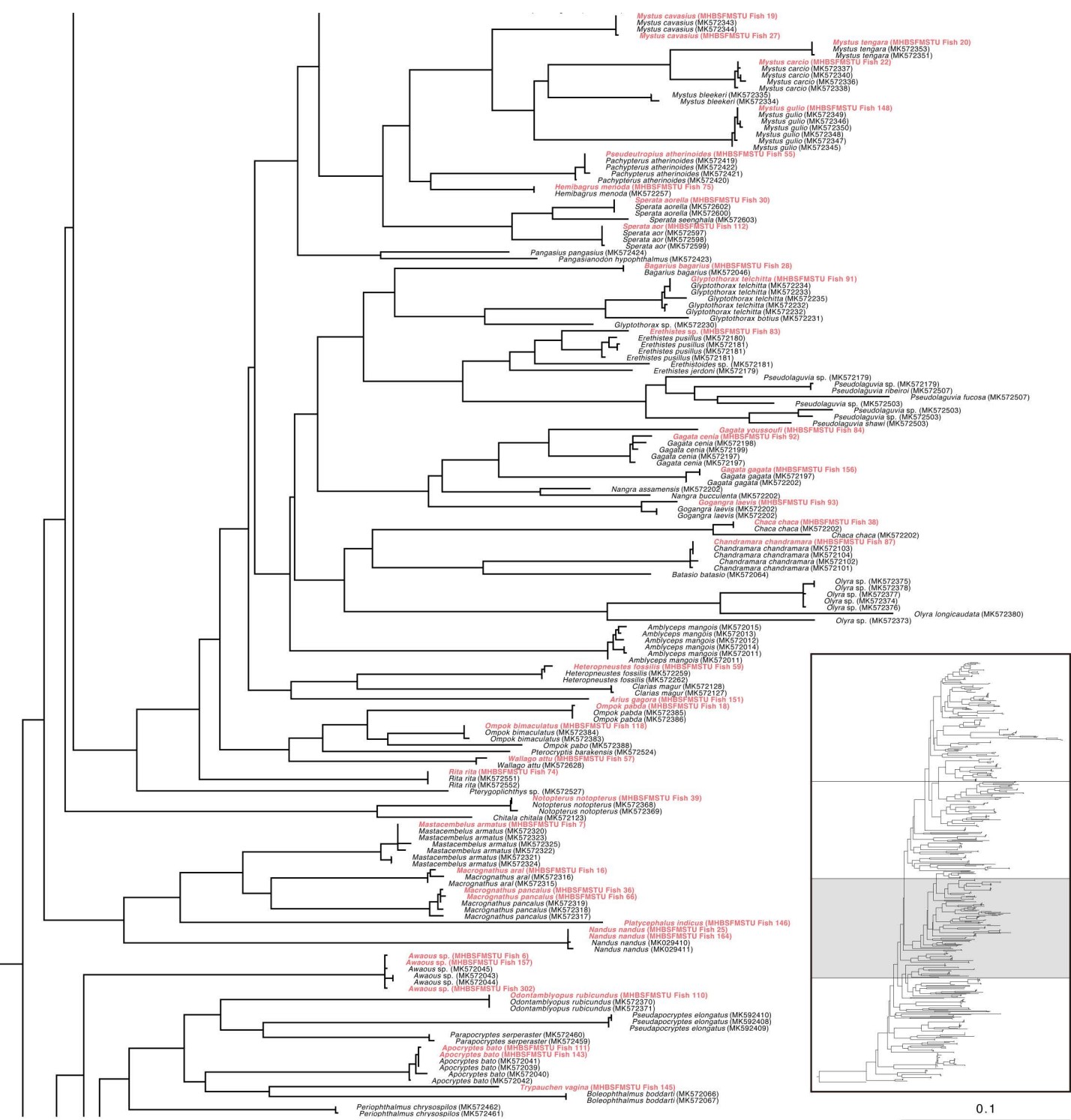

**Fig 2. Maximum Likelihood (ML) tree of freshwater species of Bangladesh based on nucleotide sequences of COI gene (partial portion of S1 Fig).**

*Erethistes*, exhibited a genetic variation of more than 5.9% from the most closely related *E. pusilus* (Fig 2). This specimen was collected from the Netrokona region. Similarly, *Schistura* sp. under the family Nemacheilidae (collected from Sakal Bazar, Jamalpur),belonging to the genus *Schistura* demonstrated a genetic variation of more than 9.2% from its closest related species *Schistura corica* (Fig 3). Two specimens of genus *Cynoglossus*, collected from Khulna and Chittagong close to the coastal belt in Bangladesh, identified as *Cynoglossus* sp. (Cynoglossidae) were found to be distantly related (COI gene divergence 17.52%) with their congeners *Cynoglossus* (MK572284) (Fig 4).The graphic quality of seven cryptic species is subpar, yet we have included images here for enhanced clarification and comprehension (**see** Fig 5).

## Discussion

Bangladesh is home to 265 [4] or 293 freshwater species, including coastal fishes, since no physiological alterations transpire during their journey [27]. Rahman et al. [14] collected 694 specimens nationwide, representing 243 species-level Operational Barcode Units (OBUs), although our sampled specimens are fewer than their findings. The absence of anticipated species in our aquatic environments was one of the problems we faced throughout the sampling procedure. The current sample size is extensive and includes the majority of available species; nonetheless, it is likely incomplete.

Our research identified seven cryptic fish species in the northeastern areas of Bangladesh and has contributed to a more thorough method of clarifying species delimitation based on COI gene. Although Rahman et al. [14] developed the DNA barcoding system, our work largely differs from their work with few overlapping aspects due to the study's inadequate coverage of Bangladesh's western and southwestern regions, primarily the Sundarbans and a fig of river Ganga and its tributaries. We attempted to cover the Sherpur region and some selected areas of the Mymensingh region as well as the exclusive region of the Old Brahmaputra River, which is included in the Ganga tributaries. Although specimens from the Khulna (Sundarban) area are limited, we have included samples from this location (see **Table 1**). Despite the several investigation based on molecular data [14,28,29], the presence of overlooked species of freshwater fishes in this study indicates that the hidden diversity among well-known and widely distributed fish species in Bangladesh might be more prevalent than previously believed. Despite being a low-lying country, Bangladesh's robust river systems impede gene flow and affect the population structure of *Hoplobatarchus tigerinus* frogs [30]. Further, both intrinsic species characteristics, such as dispersal capability and habitat specialization, and extrinsic environmental factors, such as physical obstructions and geographic separation, often determine the level of population connectivity and gene flow in the freshwater fishes [31].

The occurrence of cryptic, spatially dictated variety in fish species in Bangladesh, as indicated in this study, is not a recent revelation. Hasan et al. [2] examined the significant cryptic anuran biodiversity in Bangladesh utilizing 16S mitochondrial DNA (mtDNA); however, Raman et al. [14] contended the taxonomic ambiguity and advocated for the use of DNA barcoding (COI sequence) for the taxonomic revision of Bangladeshi freshwater fish species. We employed the COI gene in conjunction with restricted morphological data to identify the cryptic species. Identifying species merely based on physical features may prove challenging, especially when their phenotypes exhibit considerable variation [32]. Moreover, precise taxonomic identification is sometimes unattainable due to the reliance on species identification keys, which are often effective only at specific life stages. Researchers have used DNA as an alternative diagnostic tool for species, regardless of the integrated taxonomic approach [14,33]. Generally closely related vertebrate species consistently demonstrate over

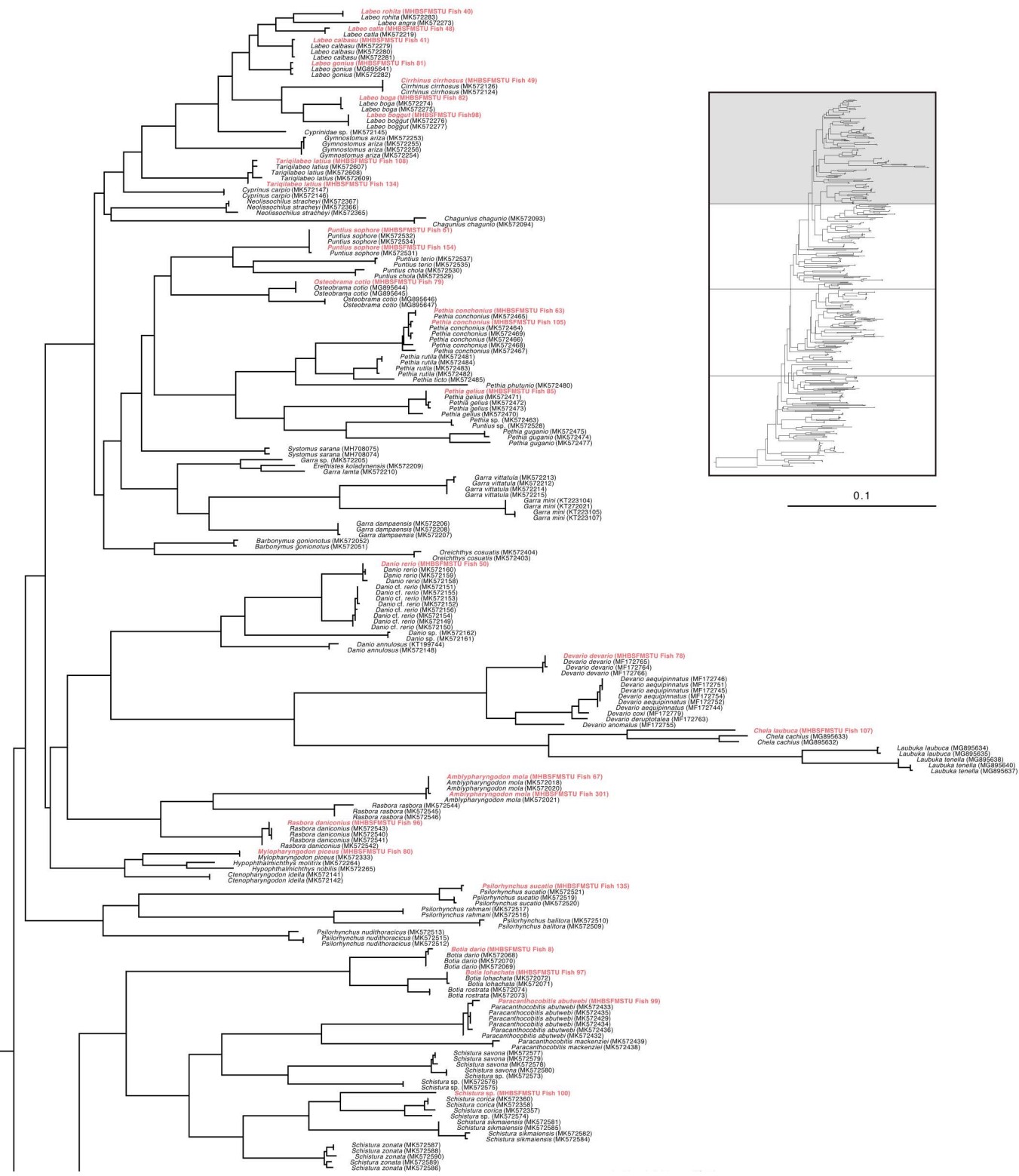

**Fig 3. Maximum Likelihood (ML) tree of freshwater species of Bangladesh based on nucleotide sequences of COI gene (partial portion of S1 Fig).**

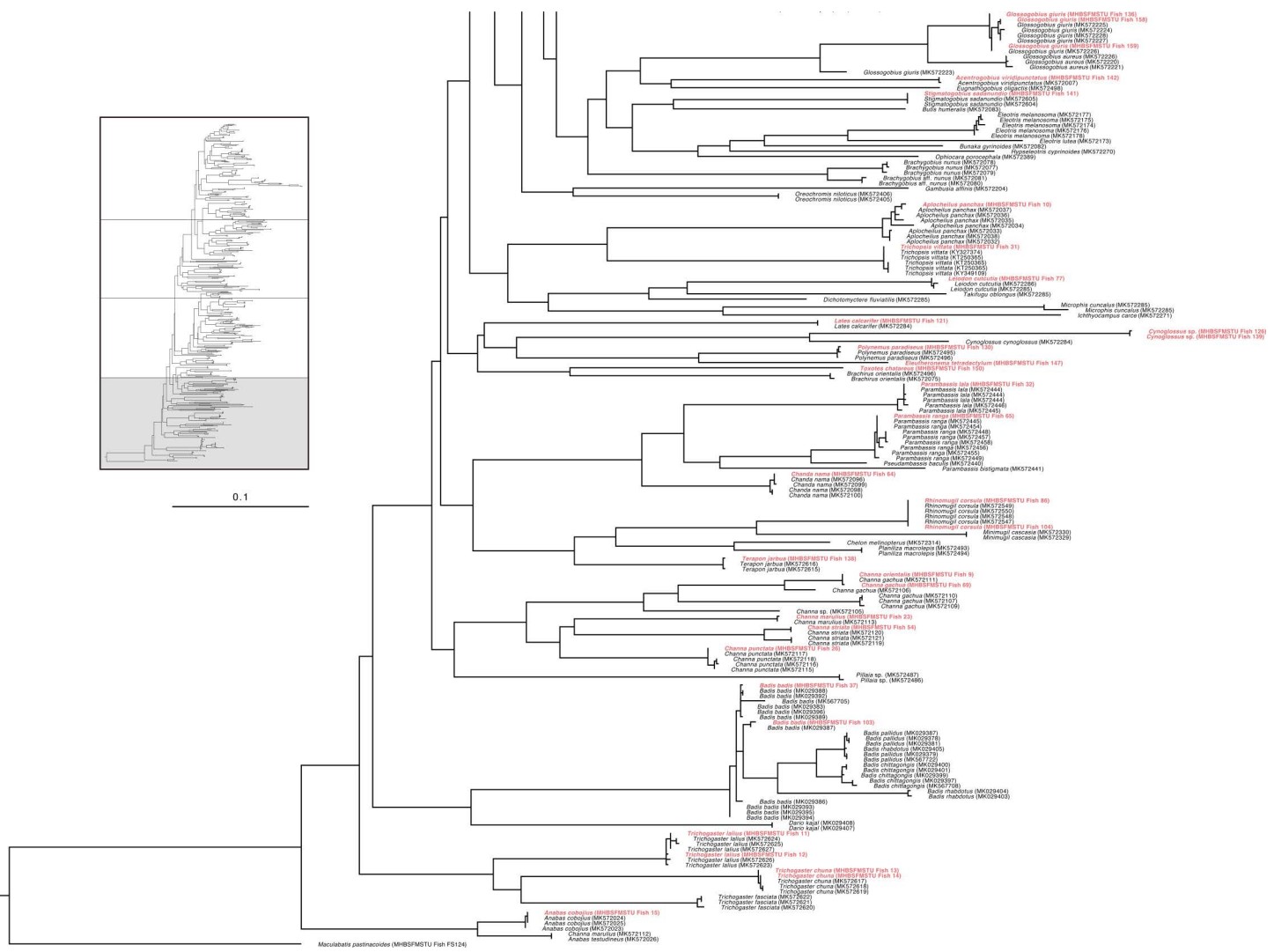

**Fig 4. Maximum Likelihood (ML) tree of freshwater species of Bangladesh based on nucleotide sequences of COI gene (partial portion of S1 Fig).**

2% divergence in the mitochondrial cytochrome b gene [34] and COX1 gene for conspecific genetic divergence in fish [35]. Fouquet et al. [36] demonstrated that a 3% cutoff value for 16S mtDNA yielded a greater number of potential neotropical frog species. In this work, intraspecific genetic divergence exceeds the minimum intraspecific distance of 3% (the COI gene), which is higher than the threshold value of 2% for fish species identification via DNA barcoding [32,37].

Based on the results of this biodiversity study, it is tempting to make conjectures about the extent of unexplored freshwater fish diversity that still exists in Bangladesh. There might still be several undiscovered cryptic species in Bangladesh. Our investigation involving the analysis of the COI gene has revealed the presence of seven likely cryptic species. These species are genetically distinct from their closely related counterparts. *Ailia* sp. is often wrongly identified as its closely related species, *Ailia coila*. Our preliminary morphological (**Table 2**) and genetic findings indicate that *Ailia* sp. is cryptic and necessitates further investigation, utilizing multiple datasets, to establish its classification as a new species. Other genera, such as *Awaous*, *Lepidocephalichthys*, *Ophichthys*, *Erethistes*, *Schistura*, and *Cynoglossus*, contain cryptic species

as well. Furthermore, the specimens we studied are poorly preserved, with some having lost their scales and/or fins. The key morphological features of each proposed cryptic species need further check and investigation with the type locality specimen whether our speculated cryptic species have any diagnosable character or not. Because, in this study we just focused cryptic species based on molecular data along with limited morphological feature which does not give any guarantee that they are each a new species. Additionally, digital photos must be captured to accurately verify the identity.

In this study, we identified several COI sequences and connected them to previously identified species, but they have recommended that the most significant aspect of the DNA barcoding is that the end users can access it and can alter taxonomies to help with identification, nomenclature, species delimitation, introgression, and multiple mitochondrial lineages [14]. However, it appears that this was only the beginning of the barcoding effort. As a result, this study is an extra effort to produce more DNA barcode data and facilitate specimen identification. A major portion of Bangladesh's small indigenous species is a primary source of protein. Together with the recent developed deep learning-based techniques [38] and molecular-morpholgy based species/genus identification [39] our DNA barcode library can assist the accurate species identification and contribute to their proper management for conservation. To prevent a potential wave of extinctions, we must quickly curtail anthropological activities that are detrimental to freshwater fish populations [40]. Immediate habitat restoration and species-specific conservation efforts are essential to save our valuable fish species and fisheries biodiversity in Bangladesh.

## Conclusion

Annotated species lists are essential for creating appropriate conservation policies. Our findings of seven cryptic species (Table 2, Fig 5) provide additional information regarding the freshwater species check list in Bangladesh. To figure out the taxonomic status of the

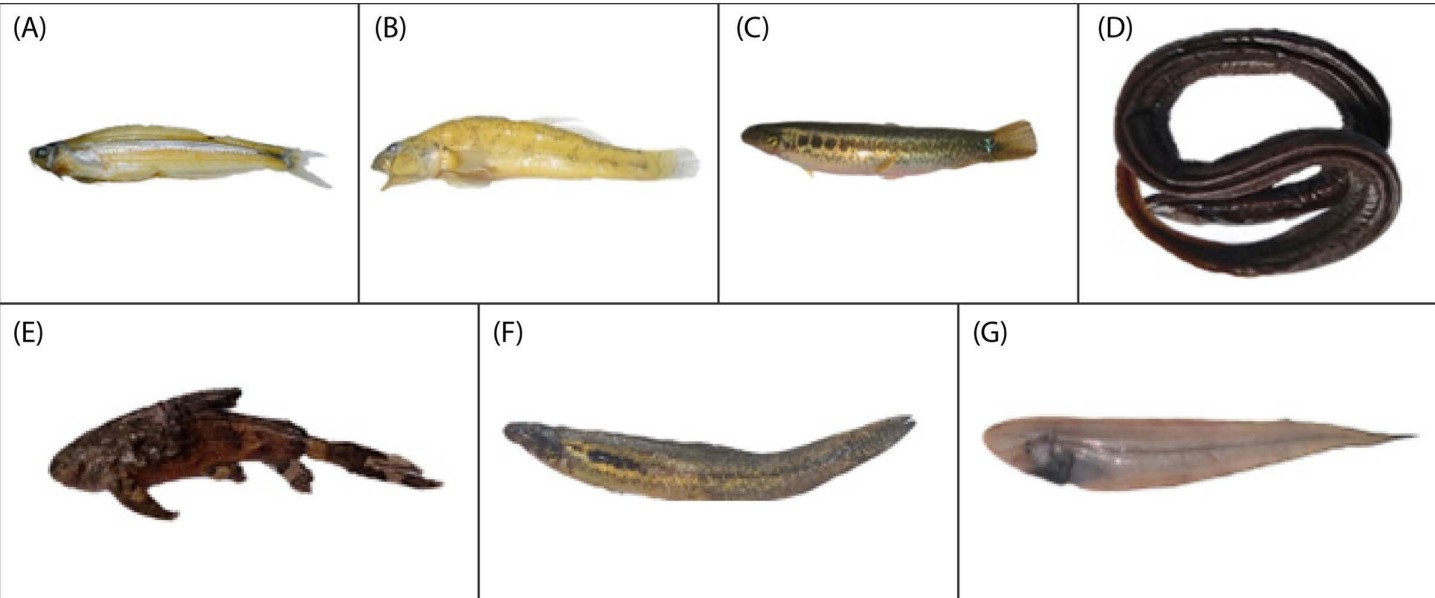

**Fig 5. Seven possible cryptic species found in this study.** (A) *Ailia* sp. (Ailiidae), (B) *Awaous* sp. (Gobiidae), (C) *Lepidocephalichthys* sp. (Cobitidae), (D) *Ophichthys* sp.(Synbranchidae), (E) *Erethistes* sp. (Sisoridae), (F) *Schistura* sp.(Nemacheilidae) and (G) *Cynoglossus* sp. (Cynoglossidae). Not in scale.

seven species that were found in this study but haven't been named yet, more comprehensive sampling that combines morphological and DNA data would need to be done. Researchers and policymakers can utilize the results of our study, which reveals that Bangladesh is home to many cryptic species, as baseline data.

## Supporting information

**S1 Fig. Maximum Likelihood (ML) tree of freshwater species of Bangladesh based on nucleotide sequences of COI gene.**
(TRE)

**S1 Table. Uncorrected *P* distance among the analyzed taxa.**
(CSV)

**S2 Table. Result of species delimitation analyses.**
(TXT)

## Acknowledgments

We are grateful to Associate Professor Dr. A. Kurabayashi, Nagahama Institute of Bio-Science and Technology for some technical support to accomplish the molecular work.

## Author contributions

**Conceptualization:** Mahmudul Hasan.

**Data curation:** Chiaki Kambayashi.

**Formal analysis:** Chiaki Kambayashi.

**Funding acquisition:** Mahmudul Hasan, Md. Saiful Islam.

**Investigation:** Mahmudul Hasan.

**Methodology:** Mahmudul Hasan, Zahid Hasan Anik.

**Supervision:** Mahmudul Hasan.

**Writing – original draft:** Mahmudul Hasan, Chiaki Kambayashi.

**Writing – review & editing:** Mahmudul Hasan, Chiaki Kambayashi.

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
