## [Decision Letter · Decision Letter 0]

14 Oct 2024

PONE-D-24-38143Cryptic biodiversity of freshwater fish species in BangladeshPLOS ONE

Dear Dr. Hasan,

Thank you for submitting your manuscript to PLOS ONE. After careful consideration, we feel that it has merit but does not fully meet PLOS ONE’s publication criteria as it currently stands. Therefore, we invite you to submit a revised version of the manuscript that addresses the points raised during the review process.

Before it be considered for publication, several issues should be fixed, which are as follows:

1.  Please provide the family name for each species following the catalog of fishes.

2.  Abstract & Introduction: need to check English texts for spelling, grammar, punctuation, and space.

3.  Materials: why the authors didn't conduct morphological analyses or morphometric analyses, which are essential in cryptic species analyses. I believe the authors should examine the most important morphological characters of fish species to identify if there are any significant differences. I suggest the authors investigate this aspect further in their study. .

4. Table arrangements: Table one is not in a good shape. For example, please provide longitude and latitude values instead of the word GPS, to be more specific, about what the authors mean by GPS.

5.  Results section: Need to create a new table that includes all the cryptic species for better clarity and harmony without providing the fish number in the text.

6. Discussion: is poor written and has to be improved based on the obtained results and referencing previous studies for comparison.

In this sense, I recommend that the MS be accepted after major revision, and I encourage its resubmission in a format that incorporates the suggestions presented here. For further information, please see the attached files and the whole comments that were made by reviewers.

We look forward to receiving your revised manuscript.

Kind regards,

Amaal Gh. Yasser, Ph.D.

Academic Editor

PLOS ONE

3. We note that your Data Availability Statement is currently as follows: [All relevant data are within the manuscript and its Supporting Information files.] Please confirm at this time whether or not your submission contains all raw data required to replicate the results of your study. Authors must share the “minimal data set” for their submission. PLOS defines the minimal data set to consist of the data required to replicate all study findings reported in the article, as well as related metadata and methods (https://journals.plos.org/plosone/s/data-availability#loc-minimal-data-set-definition). For example, authors should submit the following data: - The values behind the means, standard deviations and other measures reported; - The values used to build graphs; - The points extracted from images for analysis. Authors do not need to submit their entire data set if only a portion of the data was used in the reported study. If your submission does not contain these data, please either upload them as Supporting Information files or deposit them to a stable, public repository and provide us with the relevant URLs, DOIs, or accession numbers. For a list of recommended repositories, please see https://journals.plos.org/plosone/s/recommended-repositories. If there are ethical or legal restrictions on sharing a de-identified data set, please explain them in detail (e.g., data contain potentially sensitive information, data are owned by a third-party organization, etc.) and who has imposed them (e.g., an ethics committee). Please also provide contact information for a data access committee, ethics committee, or other institutional body to which data requests may be sent. If data are owned by a third party, please indicate how others may request data access.

Additional Editor Comments:

Before it be considered for publication, several issues should be fixed, which are as follows:

1. Please provide the family name for each species following the catalog of fishes.

2. Abstract & Introduction: need to check English texts for spelling, grammar, punctuation, and space.

3. Materials: why the authors didn't conduct morphological analyses or morphometric analyses, which are essential in cryptic species analyses. I believe the authors should examine the most important morphological characters of fish species to identify if there are any significant differences. I suggest the authors investigate this aspect further in their study. .

4. Table arrangements: Table one is not in a good shape. For example, please provide longitude and latitude values instead of the word GPS, to be more specific, about what the authors mean by GPS.

5. Results section: Need to create a new table that includes all the cryptic species for better clarity and harmony without providing the fish number in the text.

6. Discussion: is poor written and has to be improved based on the obtained results and referencing previous studies for comparison.

In this sense, I recommend that the MS be accepted after major revision, and I encourage its resubmission in a format that incorporates the suggestions presented here. For further information, please see the attached files and the whole comments that were made by reviewers.

Reviewers' comments:

Reviewer's Responses to Questions

**Comments to the Author**

1. Is the manuscript technically sound, and do the data support the conclusions?

Reviewer #1: Yes

Reviewer #2: Yes

Reviewer #3: No

2. Has the statistical analysis been performed appropriately and rigorously? 

Reviewer #1: Yes

Reviewer #2: N/A

Reviewer #3: No

3. Have the authors made all data underlying the findings in their manuscript fully available?

Reviewer #1: Yes

Reviewer #2: Yes

Reviewer #3: Yes

4. Is the manuscript presented in an intelligible fashion and written in standard English?

Reviewer #1: Yes

Reviewer #2: Yes

Reviewer #3: No

5. Review Comments to the Author

Reviewer #1: Dear Authors,

the manuscript is very interesting, explaining a problem that is emerged and is getting more and more attention. Your manuscript has some problems regarding the existence of words that have been merged between them. Please check the text for correcting them. Also I have suggested some some minor corrections to some parts of the text.

Reviewer #2: Dear Authors

I had a very close look at the MS submitted to PlosOne journal and found it interesting. However, before it be considered for publication several issues should be fixed.

1)Please use those keywords which have not been given in the title:

Ichthyodiversity, DNA barcoding, Molecular taxonomy

2) fish base is not updated data base and it is usually used for commercial or fisheries purpose.

The main acceptable data base is catalog of fishes.

You have to check the valid species name using this database:

https://researcharchive.calacademy.org/research/ichthyology/catalog/fishcatmain.asp

3) Please provide the family name for each species following catalog of fishes.

4) Table one is not in a good shape.

5) in the result section, no need to provide fish number in the text. I suggest to give all the cryptic species in a new table and provide significant findings in the result section.

6) Discussion is poor and has to be improved based on the obtained results and comparison with other related studies.

7) No conservation management text is given based on the obtained results.

8) References should be double-checked in the main text and reference section following journal's format strictly.

9) The introduction should be focused on the main and significant purpose of the ms and providing some new and update information on the cryptic species and DNA barcoding.

10) No morphological data of cryptic species has been provided. Photos of them are needed.

11) I suggest to follow the attached article published on the same subject in PlosOne. It helps to improve the ms

More are given in the attached pdf file of the Ms.

Best regards

Reviewer #3: The MS addresses the biodiversity of fish in Bangladesh, particularly intending to show that what is known today is greatly underestimated due to the methodologies used for past sampling and inventories. Although the MS brings interesting news, it does not seem to me that the sampling carried out by the authors is sufficient for an adequate analysis of the country's diversity (one of the objectives of the research), given the size of the country's drainage network and the number of specimens collected. In addition, the authors should carry out a good review of the text, correcting several cases of joined words, particularly involving species names, and improving the writing style. Furthermore, the results should clearly show the number of orders, families and species sampled and how much the survey carried out can represent in relation to the real diversity (there are no estimates that can evaluate the collection effort carried out); it is also very important to compare the results obtained with previous ones, mainly Rahman (2005) and Rahman et al. (2019). Furthermore, the authors fail to fulfill their objective "to review the biodiversity of Bangladesh's fisheries"; however, they show that the numbers may be underestimated due to the tools and techniques used previously, which were inefficient in identifying the existence of cryptic species. In this sense, I recommend that the ms be rejected in its current form, but I encourage its resubmission in a format that incorporates the suggestions presented here.

6. PLOS authors have the option to publish the peer review history of their article (what does this mean? ). If published, this will include your full peer review and any attached files.

**Do you want your identity to be public for this peer review?** For information about this choice, including consent withdrawal, please see our Privacy Policy .

Reviewer #1: No

Reviewer #2: **Yes: ** Prof. Hamid Reza Esmaeili

Reviewer #3: **Yes: ** Francisco Langeani

---

## [Author Response · Author response to Decision Letter 1]

18 Jan 2025

We did not receive any specific funding for this work. Therefore, please waive the publication fee after acceptance.

---

## [Editor Report · Decision Letter 1]

27 Jan 2025

Cryptic biodiversity of freshwater fish species in Bangladesh

PONE-D-24-38143R1

Dear Dr. Hasan,

We’re pleased to inform you that your manuscript has been judged scientifically suitable for publication and will be formally accepted for publication once it meets all outstanding technical requirements.

Kind regards,

Amaal Gh. Yasser, Ph.D.

Academic Editor

PLOS ONE

---

## [Editor Report · Acceptance letter]

PONE-D-24-38143R1

PLOS ONE

Dear Dr. Hasan,

I'm pleased to inform you that your manuscript has been deemed suitable for publication in PLOS ONE. Congratulations! Your manuscript is now being handed over to our production team.

Kind regards,

on behalf of

Dr. Amaal Gh. Yasser

Academic Editor

PLOS ONE